# Identifying correlates of Guinea worm (*Dracunculus medinensis*) infection in domestic dog populations

**Robert L. Richards**[1,2]*, **Christopher A. Cleveland**[3,4], **Richard J. Hall**[1,2,5],
**Philip Tchindebet Ouakou**[6], **Andrew W. Park**[1,2,5], **Ernesto Ruiz-Tiben**[7], **Adam Weiss**[7],
**Michael J. Yabsley**[3,4], **Vanessa O. Ezenwa**[1,2,5]

**1** Odum School of Ecology, University of Georgia, Athens, Georgia, United States of America, **2** Center for the Ecology of Infectious Diseases, University of Georgia, Athens, Georgia, United States of America, **3** Southeastern Cooperative Wildlife Disease Study, College of Veterinary Medicine, University of Georgia, Athens, Georgia, United States of America, **4** Warnell School of Forestry and Natural Resources, University of Georgia, Athens, Georgia, United States of America, **5** Department of Infectious Diseases, College of Veterinary Medicine, University of Georgia, Athens, Georgia, United States of America, **6** Chad Guinea Worm Eradication Program, Ministry of Health, N'Djamena, Chad, **7** The Carter Center, Atlanta, Georgia, United States of America

* robert.richards@uga.edu

**Data Availability Statement:** Data and code required to reproduce analyses are included in the Supporting Information files.

## Abstract

Few human infectious diseases have been driven as close to eradication as dracunculiasis, caused by the Guinea worm parasite (*Dracunculus medinensis*). The number of human cases of Guinea worm decreased from an estimated 3.5 million in 1986 to mere hundreds by the 2010s. In Chad, domestic dogs were diagnosed with Guinea worm for the first time in 2012, and the numbers of infected dogs have increased annually. The presence of the parasite in a non-human host now challenges efforts to eradicate *D. medinensis*, making it critical to understand the factors that correlate with infection in dogs. In this study, we evaluated anthropogenic and environmental factors most predictive of detection of *D. medinensis* infection in domestic dog populations in Chad. Using boosted regression tree models to identify covariates of importance for predicting *D. medinensis* infection at the village and spatial hotspot levels, while controlling for surveillance intensity, we found that the presence of infection in a village was predicted by a combination of demographic (e.g. fishing village identity, dog population size), geographic (e.g. local variation in elevation), and climatic (e.g. precipitation and temperature) factors, which differed between northern and southern villages. In contrast, the presence of a village in a spatial infection hotspot, was primarily predicted by geography and climate. Our findings suggest that factors intrinsic to individual villages are highly predictive of the detection of Guinea worm parasite presence, whereas village membership in a spatial infection hotspot is largely determined by location and climate. This study provides new insight into the landscape-scale epidemiology of a debilitating parasite and can be used to more effectively target ongoing research and possibly eradication and control efforts.

**Funding:** This study was financially supported by The Carter Center (https://www.cartercenter.org) and members of this organization were involved in the study. Ernesto Ruiz-Tiben from The Carter Center helped to conceive the study to serve the needs of the Guinea Worm Eradication Program. The Carter Center curated data which were collected by the Guinea Worm Eradication Program in coordination with the Chadian Ministry of Health. Adam Weiss and Ernesto Ruiz-Tiben from The Carter Center provided substantive comments on the manuscript prior to submission. No representatives from The Carter Center were involved in the design of the statistical methodology, statistical analysis, or preparation of the original manuscript draft. RLR was supported by a National Science Foundation Graduate Research Fellowship (https://www.nsfgrfp.org/) [grant number 2017203341]. Additional support for CAC was received from the ARCS Atlanta chapter (https://atlanta.arcsfoundation.org/). NSF and ARCS had no role in study design, data collection and analysis, decision to publish, or preparation of the manuscript.

**Competing interests:** The authors have declared that no competing interests exist.

## Author summary

The eradication of human infectious diseases has proven remarkably difficult. The world has only succeeded once, in the case of the smallpox virus. However, international efforts have driven the debilitating Guinea worm parasite closer to the brink of eradication than nearly any other parasite. Coordinated efforts by the Ministries of Health in endemic countries, the U.S. Centers for Disease Control, The Carter Center, and the World Health Organization have reduced the number of annual Guinea worm cases from millions in the 1980s to hundreds in the early 2010s, but recently a new threat has emerged. Guinea worm infections have been diagnosed in domestic dogs, particularly in the Republic of Chad, and numbers of infections have continued to increase. As in many countries where dracunculiasis is endemic, the campaign for eradication in Chad has focused intervention measures on interrupting transmission among humans, so infection in dogs jeopardizes eradication efforts. In this study, we used machine learning methods to identify demographic, geographic, and climatic factors associated with the presence of Guinea worm-infected dogs at the village level, and spatial clustering of dog cases regionally. A combination of demographic, geographic and climatic factors were important correlates of infection at the village level, but the importance of these factors varied between northern and southern populations of the parasite. At the larger village cluster level, the geographic position and climate of a village were most important. Some of our findings, including the importance of fishing villages and the difference in correlates between northern and southern villages can be used by researchers to guide additional data collection and by public health workers to better target eradication efforts. More generally, this work contributes to a broader understanding of the spatial patterning of multi-host infectious diseases of humans and animals.

## Introduction

Few human parasites and pathogens have been driven as near to the brink of eradication as the Guinea worm (*Dracunculus medinensis*). Although rarely fatal, Guinea worm disease (dracunculiasis), can be extremely painful and debilitating [1]. With a historical distribution spanning 21 countries across Asia and Africa, in the 1980s, Guinea worm was initially targeted for eradication by the World Health Assembly [2]. As a result, the number of Guinea worm disease cases per year decreased from 3.5 million to hundreds between the mid-1980s and the early-2010s [3]. However, this consistent downward trajectory slowed in a handful of endemic countries by the late 2000s. For example, in Chad, after an eradication campaign from 1993 to 2000, there were no Guinea worm cases reported for 10 consecutive years (2000–2009), but then there was a small, unexpected, outbreak in 2010 [4]. Since then, disease cases have persisted and all available evidence links the persistence of human cases to the emergence of Guinea worm in domestic dogs (*Canis lupus familiaris*) [2,3,5,6].

*D. medinensis* infection classically occurs through the ingestion of infected cyclopoid copepods [7]. The transmission cycle begins when, after an approximately 10–14 month incubation period, a female worm emerges, often on the feet or legs of a host, and releases first-stage larvae into a water source. Guinea worm larvae are ingested by copepods where they undergo development to an infectious third-stage larva [4]. Typically, ingestion of infectious copepods occurs when mammalian hosts drink unfiltered water, but recent work suggests that transmission may also occur when humans or other mammalian hosts (including dogs, cats, and baboons) eat undercooked or raw fish or frogs [3,4,8,9]. These aquatic animals are known

predators of copepods potentially allowing them to act as paratenic or transport hosts in transmission to mammals. Given the Guinea worm transmission cycle, classical eradication efforts have primarily involved strategies to prevent human ingestion of infectious copepods (e.g. use of safe water sources such as borehole wells, filtering potentially-contaminated water, chemical treatment of surface water), or to limit larval shedding by emerged female worms (e.g. incentives for reporting cases, containment of infected individuals) [7]. Many of the same control strategies applied to human disease are also now used to control Guinea worm transmission in dogs, along with new approaches such as recommendations to bury or burn fish entrails to prevent dog ingestion of the parasite [2]. However, while these efforts have been largely effective at interrupting human transmission, they have not influenced numbers of dog infections in the countries where the parasite remains endemic [10]. Thus, additional insight into the correlates of Guinea worm transmission is needed to help inform future elimination strategies for dogs.

The dependence of the Guinea worm life cycle on the environmental availability of water provides an intuitive starting point for exploring potential correlates of Guinea worm infections in dogs. In Chad, the endemic region for Guinea worm occurs in a riparian floodplain along the Chari River where fishing is an essential form of subsistence and commerce. Insight into Guinea worm transmission in dogs might therefore be gained by examining associations between disease patterns and demographic (e.g. fishing practices of a village), climatic (e.g. rainfall), and geographic (e.g. proximity to water sources) features that are relevant to the transmission process. Indeed, such approaches have been used to inform control efforts for other environmentally-dependent diseases [11–13]. For example, spatial modeling of schistosomiasis risk has been used to identify correlates of risk and to target control efforts such as mass drug administration, alleviating the need for costly long-term monitoring [14,15].

In this study, we investigated potential factors associated with detection of Guinea worm infection in domestic dogs in Chad and the implications for disease elimination. We used a machine learning approach to identify the demographic, geographic, and climatic factors that explain variation in Guinea worm infection in over 2000 villages at two different spatial scales: the village scale and the multi-village spatial hotspot scale. The hotspots represent areas with significantly more dog infections than expected given the underlying size of the dog population. Overall, by providing information on factors that predispose villages for canine Guinea worm infection, our study can help guide ongoing research, surveillance, and potentially elimination efforts.

## Methods

### Infection presence and predictor variables

We obtained data on *D. medinensis* infections in domestic dogs from the Guinea worm eradication program led by the Chad Ministry of Health and supported by The Carter Center and the World Health Organization [10]. The surveillance data used in this study were collected between 2013–2017 from 2125 villages located along the Chari river. Surveillance involves regular searches of households for humans or animals showing signs of Guinea worm infection as well as a system of incentives for case reporting by community members. Infections are only recorded when a *D. medinensis* worm emerges from the skin, becoming visible to Guinea Worm Eradication Program (GWEP) staff, thus an infection report reflects the definitive occurrence of a *D. medinensis* infection by a single worm. Over the five-year study period, the village-level prevalence (proportion of villages with an infection) of *D. medinensis* in dogs was 19.1% (406/2125), but the number of infections per village varied widely (e.g. 2016 range: 0–71), and generally increased over time. In 2013, there were 57 infections in 39 villages, but

this number increased to 1287 infections in 238 villages by 2016. This increase in infections is likely the result of both the spread of the parasite and increased intensity of surveillance [10]. For our analyses, we considered a village to be positive for Guinea worm if there had been at least one dog infection reported at any time during our five-year study window. This approach allowed us to minimize the effect of increased surveillance effort over time and to better align infection data with the temporal resolution of environmental predictor variables. Our approach also better matches the needs of the GWEP to identify currently uninfected villages likely to become infected. For example, climatological data (see below) are based on a time-averaged interpolation from 1970–2000. Finally, given recent work on the genetic structure of Guinea worm in Chad identifying Northwestern and Southeastern parasite sub-populations based on a clear spatial division of genetic relatedness ([6], Fig 1), we also accounted for spatial structure of the parasite population in our analytical approach (see Data Analysis).

To understand potential correlates of *D. medinensis* infection patterns in dogs, we collated data on 36 variables that might contribute to transmission and 2 variables estimating surveillance intensity. All predictor variables were collated at the village level and fell into one of four general categories: demographic, climatic, geographic, and surveillance (see S1 Table). Demographic variables, such as dog and human population size estimates, were collected by GWEP staff during house-to-house surveys conducted each January and a mean value, for the five-year study period, was calculated for each village. These population-based variables were included in our model because of well-described positive associations between host population size and parasite transmission in other systems [16–18], and as a method to control for the increased probability of parasite detection in larger, more populated areas. The identity of a village as a 'fishing village' was of particular interest because of the possible connection between unattended fish scraps and *D. medinensis* transmission to dogs [8]. Villages were identified by GWEP staff as "fishing" if greater than 50% of families in that village fish.

Climatological variables, including variables related to temperature and precipitation, were included as predictors in our model because of the relevance of ephemeral and seasonal water sources to the life cycle of the *D. medinensis* [1]. We conducted all analyses with both high spatial resolution climate data from the WorldClim dataset [19](primary analysis) and with lower spatial resolution data collected via remote sensing over the same time period as the parasite observations (alternative analysis). Remotely sensed temperature data were derived from the Modern-Era Retrospective Analysis for Research and Applications, version 2 (MERRA-2) dataset [20,21], while precipitation data were obtained from the African Rainfall Climatology, version 2 dataset (ARC2) [22](See S1 Table). Model results based on the primary analysis with WorldClim data are presented in the main text but differences between results of the World-Clim and alternative remote sensing analyses are noted where relevant.

Geographic variables such as land-cover type (e.g. vegetation type and proportion of cover), remotely sensed surface water, and elevation were included in our model because they might dictate the location, size, and permanence of water sources that are crucial to the Guinea worm parasite life cycle. Both climatic and geographic covariates were extracted for villages according to latitude and longitude coordinates. GWEP workers collected these coordinates at the center of each village using portable GPS devices.

Finally, to control for spatial variation in surveillance, we included two measures of surveillance effort in our models, the mean number of healthcare workers present in a village annually and the total number of healthcare supervisor visits to a village from 2013–2017. These two metrics capture different components of surveillance intensity: the first variable summarizes spatial variation in average surveillance intensity over the study period, while the second variable encompasses some degree of the spatiotemporal variation in surveillance since villages have no visits for years during which they were unsurveilled.

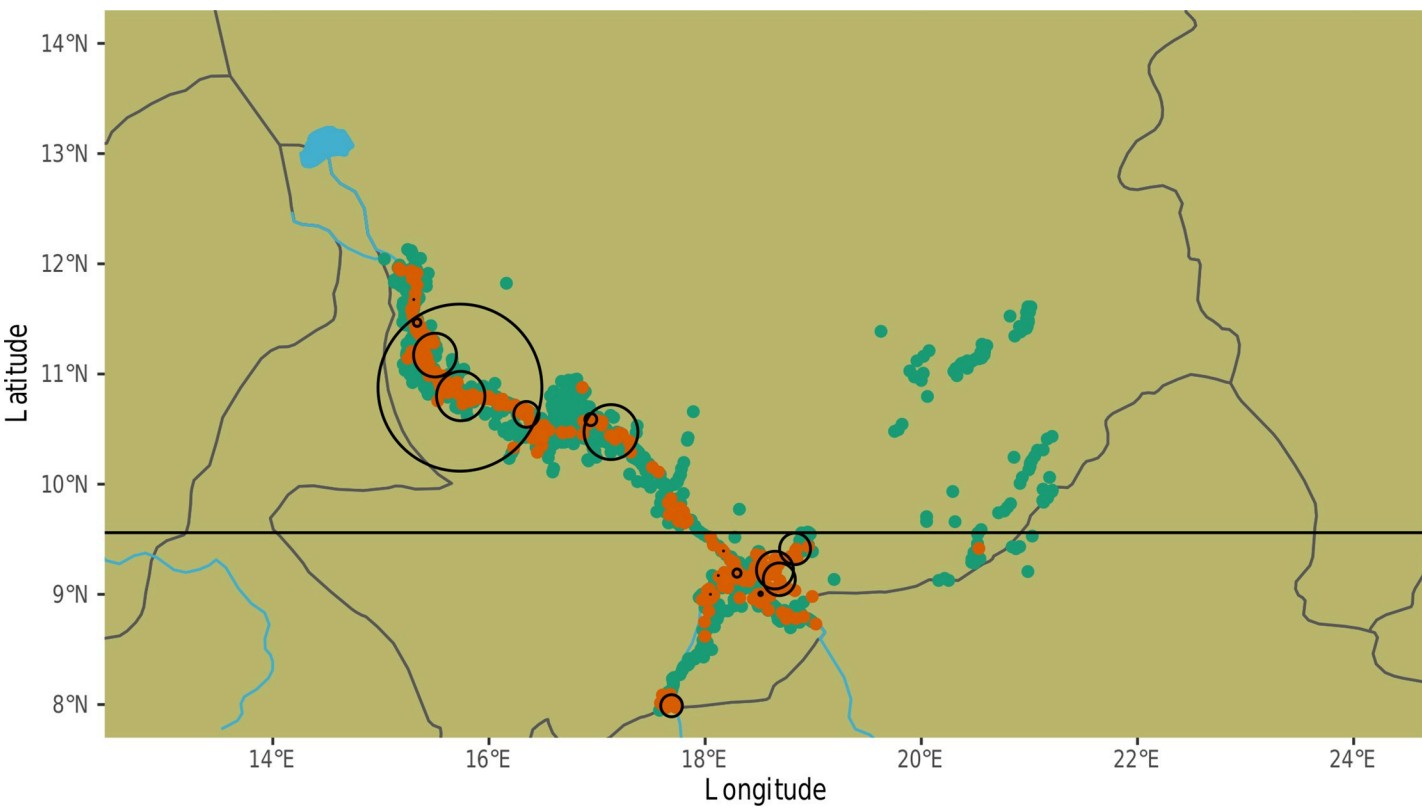

**Fig 1. Map of spatial hotspots of *D. medinensis* infection in dogs.** Orange points represent villages where an infection was present (n = 312), while green points are villages without a history of dog infection (n = 1280). Circles represent spatial hotspots identified by hotspot analysis. The horizontal line marks the rough geographic delineation between Northern and Southern subpopulations of the parasite.

### Data analysis

**Spatial hotspot analysis.** We used spatial scan analyses to identify whether 'hotspots' of *D. medinensis* infections in dogs exist (i.e. whether villages with dog cases are spatially clustered). Prior to identifying hotspots, all villages with missing records for dog population size (N = 135) were removed leaving 1990 villages in the hotspot analysis, including 1592 in the training set and 398 in the held-out evaluation set. The spatial analysis specified a discrete Poisson incidence model which assumes that the mean number of detected dog infections in a village is proportional to the population of dogs in that village and conducts a spatial scan to identify areas with significantly more infections than expected. Spatial cluster analyses were performed using the *rsatscan* package [23] in R version 3.5.1. Because the spatial scan is performed at varying radiuses centered on all villages, overlapping or concentric clusters are possible [24]. We classified villages based on whether or not they were members of a spatial infection hotspot.

**Boosted regression tree analysis.** We used boosted regression trees (BRTs) to identify variables (see S1 Table) of importance for predicting *D. medinensis* infection in dogs at the local (village) and regional (hotspot) scales. This method has been widely used both for the predictive modeling of the spatial extent of infectious disease risk [25–27] and for the identification of important predictors of risk [11–13].

Prior to including all variables in our boosted regression tree models we diagnosed significant co-linearity in all types of predictors using Ward-clustering based on the spearman

correlation matrix [28,29]. Co-linear clusters were reduced to the single most central variable in the cluster (the variable with highest mean correlation to all others) according to [28]. After co-linearity reduction, villages were randomly sub-sampled into training (80%, 1700 villages) and testing (20%, 425 villages) sets for the purpose of final model validation. Due to constraints of the hotspot analysis (villages without records of dog population could not be included in the model) we only included 1592 (80%) and 398 (20%) villages in the training and testing set respectively for the boosted regression tree trained to hotspot identity.

Our BRT models identified variables predicting: (i) whether *D. medinensis* was detected in a village over the five-year study period and (ii) whether a village was a member of an infection hotspot (note that membership in an infection hotspot was not limited to villages with previous infections since uninfected villages in close proximity to villages with large numbers of cases can be identified as members of a hotspot). To evaluate any potential effects of parasite genetic structure, we also re-fit models using data exclusively from the northern/western and southern/eastern parasite sub-populations respectively. All models were fit with the *gbm.step* function from the *dismo* package [30] in R version 3.5.1 which iteratively fits boosted regression tree models with increasing numbers of trees and assesses the fit of the model through 10-fold cross validation. The tree-number that minimizes the residual deviance in the response variable was chosen as the best model. Models were assessed for their fit through cross-validation and to held-out evaluation data using area under the receiver operator curve (AUC), a measure of a model's ability to discriminate between positive and negative responses [31]. Model parameters (bag fraction, learning rate, and tree-complexity) were tuned to maximize AUC through cross validation while minimizing the difference between training and cross-validation AUC. This served as a control on overfitting to training data. Bag fraction denotes the size of the random sample on which each tree is trained, learning rate sets the shrinkage of the contribution of a given tree to the overall model, and tree-complexity defines the number of branch splits allowed in the tree. Tuning for tree complexity requires fitting interactions between covariates. The strength of the interactions and a partial dependence plot of the strongest interaction for the presence-absence (S3 Table, S4 Fig) and hotspot (S4 Table, S5 Fig) models are reported in the supplementary material since they do not provide information that qualitatively changes the interpretation of the non-interaction based results. The relative importance of each covariate was determined by calculating the relative improvement in model fit when a covariate is included in the model, weighted by how often the covariate appeared in the collection of trees [32]. Values were then rescaled such that they summed to 100, with larger values representing a larger relative influence on model fit [33,34]. We visualized the effect of each covariate on the response variable using partial dependence plots [33,34].

## Results

Over the five-year study period, the village-level prevalence (proportion of villages with an infection) of *D. medinensis* in dogs was 19.1% (406/2125), but the number of infections per village varied widely (e.g. 2016 range: 0–71), and generally increased over time.

### Identifying spatial hotspots of infection

Seventeen spatial hotspots of *D. medinensis* infection were identified with a p-value < 0.05. Overall, 43.1% (857 out of 1990) of villages were identified as belonging to a spatial hotspot (Fig 1). 29.5% (253/857) of hotspot villages actually had a detected case during the five-year study period. Hotspot villages were located primarily along the northern/western reaches of the Chari river (88.1% of hotspot villages are in the North compared to 74.9% of all villages),

with smaller clusters in the southern/eastern reaches (11.9% of hotspot villages are in the South compared to 25.1% of all villages). The total number of infections was, however, relatively evenly split between northern/western (1361) and southern/eastern (1195) villages, suggesting that cases are more spatially concentrated in the southern/eastern region.

### Evaluating predictors of infection at different scales

Two clusters of predictor variables were identified from our co-linearity analysis (S2 Table). The first co-linear cluster contained longitude, latitude, mean elevation, and the majority of the bioclimatic variables, and this cluster was reduced to the one central variable, mean annual precipitation (Bioclim 12). The second co-linear cluster contained only human population and number of households in a village and was reduced to human population size. The BRT model fit data on *D. medinensis* presence across all villages with a mean AUC of 0.887 in cross-validation and an evaluation AUC of 0.889. Demographic and geographic variables were the most important predictors of *D. medinensis* presence across all villages, with dog population size (22.2% Importance; Fig 2A, Fig 3A), number of healthcare supervisor visits (17.8% Importance; Fig 2A, Fig 3B) identity as a fishing village (14.6% Importance; Fig 2A, Fig 3C), standard deviation in elevation (11.1% Importance; Fig 2A, Fig 3D), and mean annual precipitation (Bioclim 12; representing covariate cluster 1, 9.4% Importance; Fig 2A) emerging as key predictor variables of importance. Splitting villages into northern (CV AUC: 0.905, evaluation AUC: 0.915) and southern (CV AUC: 0.826, evaluation AUC: 0.810) parasite sub-populations, revealed strong distinctions between the two regions. Fishing village identity and standard deviation in elevation were important in northern villages, while the importance of climate-related variables, represented by mean annual precipitation (Bioclim 12; cluster 1) and mean temperature of the driest quarter (Bioclim 9) only emerged in southern villages (Fig 2B and 2C, S1 Fig, S2 Fig).

When fit to spatial hotspot identity, the BRT model achieved a mean AUC in cross validation of 0.988 with an evaluation AUC of 0.989. The most important variables included cluster 1, represented by mean annual precipitation (Bioclim 12, 36.0% Importance; Fig 2D, Fig 4A), number of healthcare supervisor visits (26.0% Importance; Fig 2D, Fig 4B), mean temperature of the coldest quarter (Bioclim 11, 10.2% Importance; Fig 2D, Fig 4C) and mean temperature of the driest quarter (Bioclim 9, 9.6% Importance; Fig 2D, Fig 4D). These findings were not qualitatively different when analyses were performed on northern and southern villages separately (S3 Fig).

Finally, results at the village and scale were fairly robust to variation between our two climate data sources, the WorldClim data used in our primary analysis and the contemporary, lower spatial resolution, remotely sensed data used in an alternative analysis. Climate related variables were relatively less influential in models in the alternative analysis, possibly because of the coarser spatial resolution of these data. A lack of explanatory power of climate variables in the alternative analysis emerged in the hotspot analysis as well. Remotely sensed population density (12.9% Importance; S6 Fig, S8 Fig) and standard deviation in elevation (7.6% Importance; S6 Fig, S8 Fig) increased in importance while the only climatic variable retained among the top variables was precipitation of the warmest quarter (Bioclim 18, 11.7% Importance; S6 Fig, S8 Fig). Precipitation of the warmest quarter was one of the variables contained in the highly important cluster 1 which emerged as important in our primary analysis with WorldClim data (S2 Table).

## Discussion

We found that a combination of demographic, geographic, climatic, and surveillance variables were important predictors of detection of Guinea worm infection in dogs in the villages of

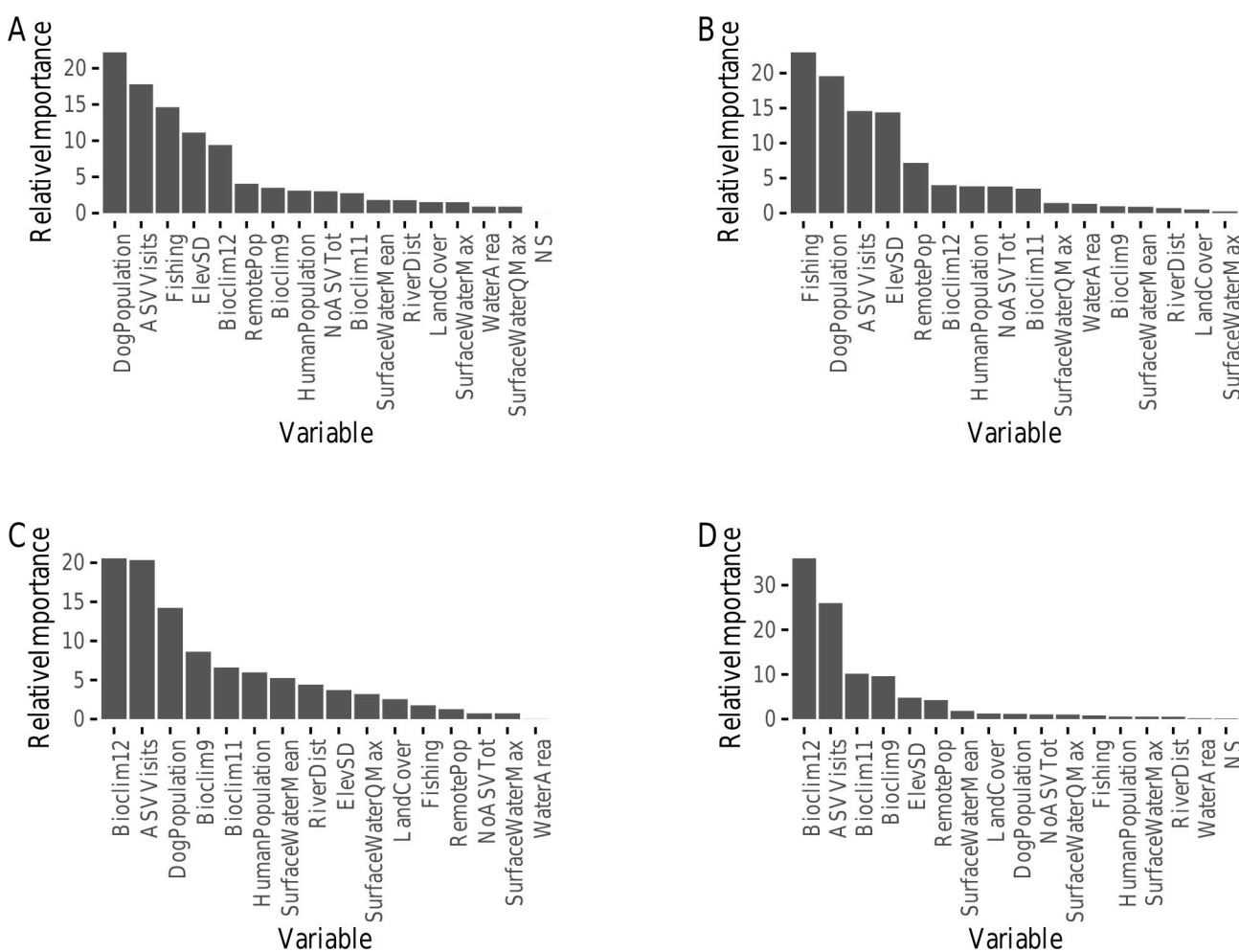

**Fig 2. Relative importance estimates.** Relative importance of covariates in predicting (a) *D. medinensis* infection presence in a village, (b) infection presence in Northern villages, (c) infection presence in Southern villages, and (d) hotspot identity across all villages. See S1 Table for notes on variable abbreviations (e.g. ASV Visits corresponds to the total number of healthcare supervisor visits to a village from 2013 to 2017).

Chad. The surveillance variable of healthcare supervisor visits was an important predictor of Guinea worm infection at both the individual village and spatial hotspot scales. The size of the dog population and the status of a village as a fishing village were demographic predictors of Guinea worm infection at the village scale, while standard deviation in elevation emerged as a geographic predictor at this scale. At the hotspot scale, climate and geography were paramount. Overall, our results suggest that some demographic and geographic features of villages (e.g. fishing village identity and variation in elevation) can be used to inform local Guinea worm control efforts. In contrast, broad scale hotspots of infection are largely determined by climate or geographical location suggesting that spatial epidemiology may be most appropriate for identifying infection risks at this broader scale.

At the smaller scale of our analysis (the village), our models identified dog population size, fishing village identity, and number of visits from healthcare supervisors as key variables predicting presence of *D. medinensis* infection in dogs in a village. The importance of dog population size to parasite infection at the village scale is unsurprising, but the mechanisms behind this relationship are uncertain. Larger dog populations represent more chances that an infected copepod in the environment will encounter a viable host but not necessarily a larger

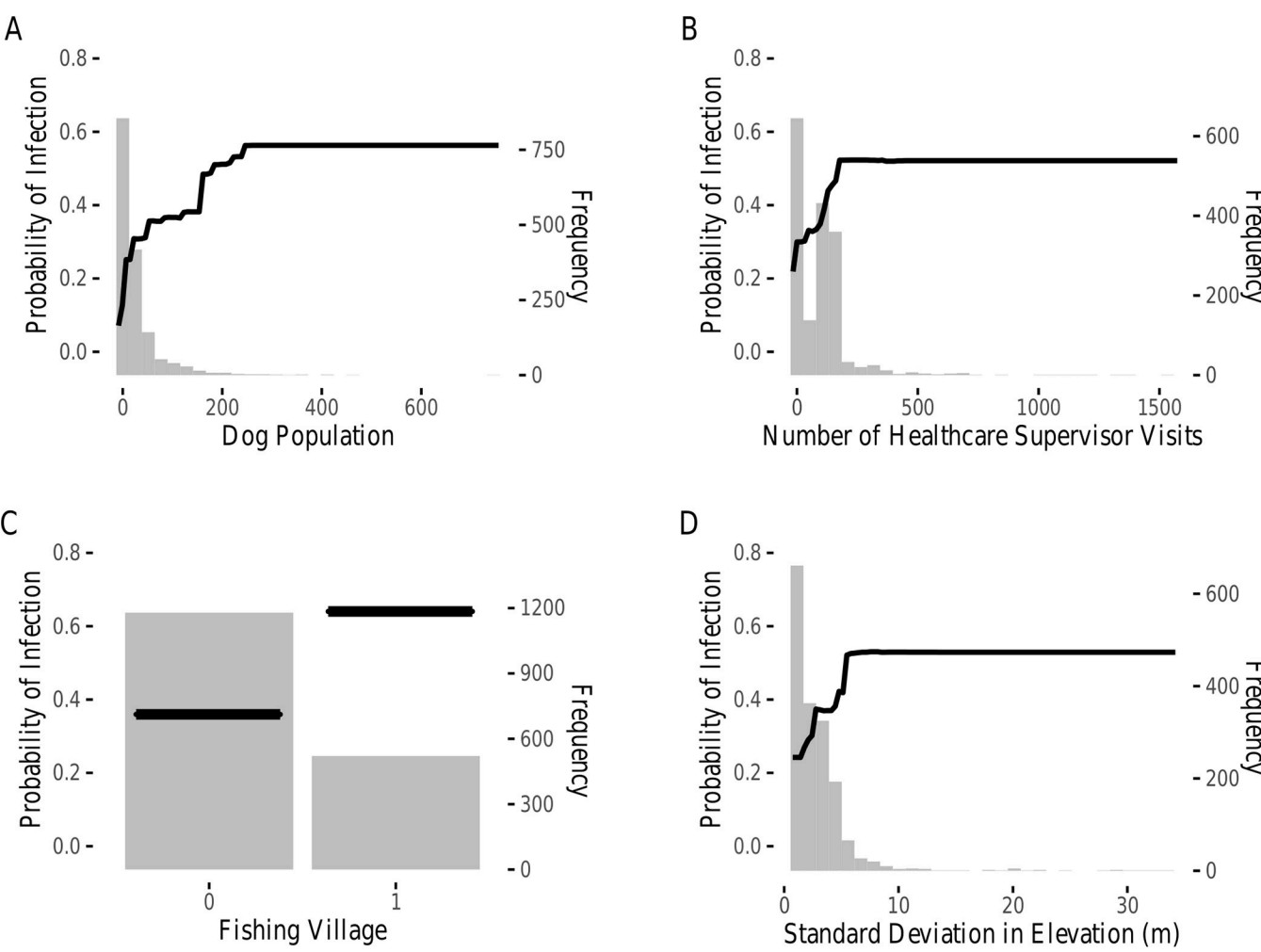

**Fig 3. Village-level partial dependence plots.** Partial dependence plots showing the effect of (a) dog population, (b) number of healthcare supervisor visits, (c) identity as a fishing village, and (d) standard deviation in elevation on probability of dog infection. Histograms represent the distribution of values for these covariates amongst all training villages.

risk to an individual dog or human. A larger dog population may also simply increase the probability that a Guinea worm infection is detected. In this case, dog population may be an indicator of surveillance effort, similar to healthcare worker surveillance. Unfortunately, the temporal scale of our analysis prevents us from drawing strong conclusions about the mechanisms underlying this pattern. Future work based on finer-scale temporal data is needed to better understand the role of dog population size as a potential driver of Guinea worm infection risk.

The importance of fishing village identity in driving infection risk at the village scale was also notable. This finding corroborates the evolving understanding of the possible role of fish in the transmission of Guinea worm. Because of the predominance of fishing activity in fishing villages, these are the locations where dogs are most likely to have access to raw fish and fish entrails which may serve as transport hosts of the parasite [8,35]. Fishing villages may also have other features in common besides the presence of fish remains, however, we account for many such features in our models by including variables such as distance to permanent water, surface water area, longitude, latitude, human and dog population sizes, and surveillance by

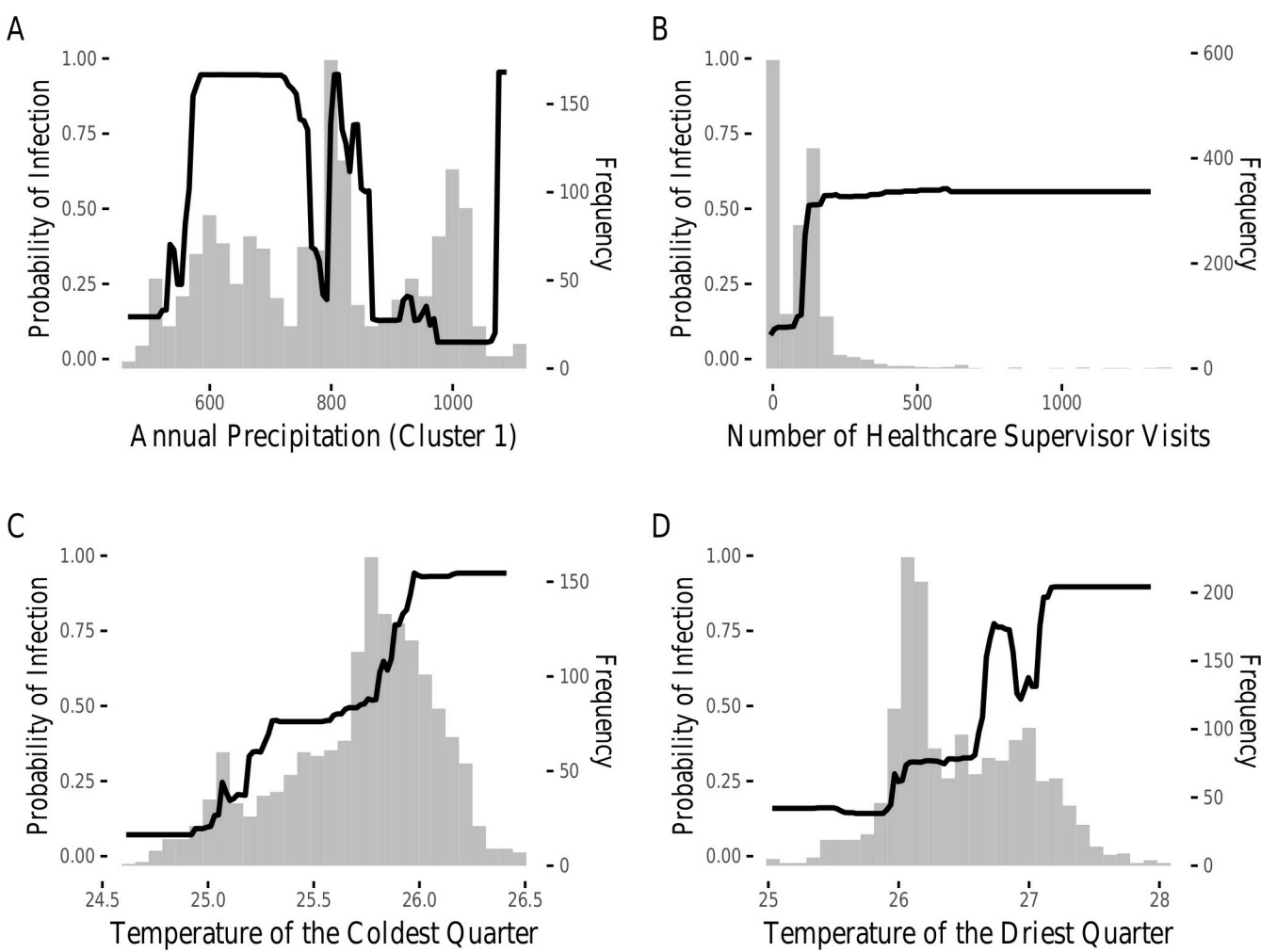

**Fig 4. Hotspot-level partial dependence plot.** Partial dependence plots showing the effect of (a) cluster 1 (represented by annual precipitation [Bioclim 12]), (b) number of healthcare supervisor visits, (c) temperature of the coldest quarter [Bioclim 11], and (d) temperature of the driest quarter [Bioclim 9] on probability of dog infection. Histograms represent the distribution of values for these covariates amongst all training villages.

healthcare workers. Intriguingly, increased infection risk among fishermen or fishing cultures has been found in other freshwater-associated parasitic diseases, like schistosomiasis [36]. However, the elevated risk for schistosomiasis is likely due to increases in high risk behaviors, like wading, rather than to proximity to water *per se* [37,38]. Our parallel finding that a fishing culture is associated with higher risk of Guinea worm infection in the dogs of a village suggests that addressing human behaviors that facilitate transmission through fishing-related activities may enhance disease control. However, future work is needed to understand the specific human behaviors captured by the 'fishing village' designation that translate to increased infection risk in dogs.

The standard deviation of elevation also emerged as a highly important variable at the village level. Variation in elevation may contribute to a variety of factors influencing parasite transmission, but the most likely of these is that increasing variation in elevation increases the number and size of areas where ephemeral water sources can form, either after rainfall events or as flood waters recede. This idea is supported by the fact that the elevation data used in our models (NASA Shuttle Radar Topography Mission based Digital Elevation Model data) are

commonly employed in disaster management and flood projection with the assumption that low elevation areas on a landscape are most likely to flood first and recede last [39–41]. These data have also been regularly used to explain infection risk for other water associated diseases such as West Nile virus and malaria [42,43]. Importantly, ephemeral pools may serve as water sources for dogs, possibly concentrating infected copepods and increasing the likelihood of Guinea worm transmission [4].

Intriguingly, the relative importance of fishing village identity and standard deviation in elevation for predicting Guinea worm infection risk varied regionally. The Guinea worm population in Chad forms two distinct genetic sub-populations, one northwest of Manda National Park and one southeast of the park [6]. Our work suggests that dog exposure risk in these two sub-populations is driven by different factors. While dog population size and the number of visits from healthcare supervisors were important in all regions, the difference between fishing villages and non-fishing villages was smaller in southern/eastern villages. The downgrading of fishing village identity as a key predictor variable in the south may suggest differences in the predominant mode of parasite transmission between the two regions. In particular, the two regions likely vary in the relative importance of consumption of fish scraps as a mechanism of transmission. Additionally, the emergence of key climate-related variables for southern villages in our analysis of southern vs. northern villages, and the downgrading of standard deviation in elevation (Fig 2, S1 Fig, S2 Fig), suggests that precipitation may be more important than topography for facilitating the type of dog-water contacts that promote parasite transmission in the southern region. Indeed, the tendency for the drivers of human and animal parasites to differ regionally due to social and environmental differences has been described in other systems [44,45]. These differences can pose significant challenges for disease control, in part, because they generate cryptic heterogeneity in the efficacy of control measures [46–48]. For Guinea worm in Chad, our findings suggest that control strategies might need to be tailored by region to maximize efficacy.

In addition to our village level analyses, the identification of spatial hotspots allowed us to investigate the correlates of larger scale patterns of infection. At this broader scale, covariates which vary significantly between nearby villages (e.g. demographic variables like dog population size) provided minimal discriminatory power, with the notable exception of healthcare supervisor visits. Instead, climatic and geographic factors accounted for 55.8% of total variable importance. The fact that the cluster 1 variable, represented by mean annual precipitation and reflecting key geographic and climatic factors, was a more important predictor of infection presence at the hotspot compared to village scale may, in part, be the result of some important village-level variables (e.g. dog population size, fishing village identity) becoming obscured at the hotspot scale. The inclusion of uninfected villages into spatial hotspots may explain this effect. At the hotspot scale, climatic and geographic variables, which vary gradually across space, are likely to show more consistency across a hotspot than demographic variables, thereby serving as more reliable predictors of hotspots. However, the fact that mean annual precipitation clusters with so many climatic and geographic variables makes it difficult to ascertain exactly what factors drive hotspot risk beyond the strong predictive importance of climate and geography. Alternatively, the location of these large, high infection, hotspots may be driven by spatial infection processes including the flow of infected humans and dogs between adjacent villages. In this case, climatic and geographic variables may simply serve as proxies for the importance of proximity to certain high-risk areas. This effect may be particularly relevant in our model due to the very large infection hotspot identified in the north-west Chari region (Fig 1), which likely dominated the boosted regression tree analysis. Alternative modeling strategies which explicitly consider transmission paths between villages and hotspots

should be used to generate more nuanced predictions about the spatial drivers of infection risk [18,49].

There are a few important caveats that should to be considered alongside our findings. First, we aggregated our data across a five-year period, assuming no change in the correlates of presence or absence of Guinea worm across the entire study period. This assumption was necessary given the structure of our data but may prove misleading if changes in predictors over the study period were associated with a rapid rise in numbers of cases. Second, we explicitly included seasonality only for the climatic and surface water variables in our models. However, it is likely that other variables, such as dog and human population sizes, also vary seasonally in ways that could influence the incidence of *D. medinensis* [50]. In this case, the value reported for these predictors in our models may not represent the period for which the effect of these predictors is most pronounced. This issue could have obscured the importance of demographic covariates in our models. Finally, Guinea worm surveillance effort in Chad varied both spatially and temporally during the study period, which may have complicated our analysis. We limited the effect of temporal variation in surveillance in our presence-absence analysis by analyzing data aggregated across multiple surveillance years. We also controlled for spatial variation by including multiple measures of surveillance effort in our models. Importantly, these surveillance covariates accounted for a substantial amount of variation in parasite presence and hotspot identity, suggesting that the remaining variation between villages may be largely independent of differences in surveillance intensity. Despite these limitations, the results of our study align with the general biological understanding of Guinea worm and contribute new information on the correlates of transmission of this parasite.

Current efforts to eradicate Guinea worm disease as a human health concern will need to deal with the presence of a new animal host: the domestic dog. This includes efforts already underway to elucidate new pathways involved in the parasite life cycle (e.g. [4,8,9,35]) and to clarify the modes of indirect transmission between dogs and humans (e.g. [6]). In this study, we approach the problem from a different, but synergistic, perspective. We used prior knowledge of the natural history and epidemiology of the system to generate a set of variables that might influence transmission and asked which are most important for predicting the detection of *D. medinensis* infection in villages across Chad. Our results support the epidemiological importance of demographic and geographic factors such as human fishing and elevation, at the local scale in northern villages, while emphasizing the importance of climatic correlates in southern villages. In aggregate, our study suggests that localized control measures at the village level, such as the treatment of water sources, should be targeted based on demographic and geographic factors in the north and geographic and climatic risk factors in the south. In contrast, regional control measures, like public information campaigns, should be targeted based on known geographic risk areas. In particular, future work should aim to validate our models with additional surveillance and experimental data to uncover the mechanisms linking Guinea worm transmission to key predictor variables identified in this study.

## Supporting information

**S1 Table. Model covariates.** A list of the source, resolution, and means (+/- SE), where appropriate, for all variables included in the boosted regression tree models. Climatic, surface water, and floodwater summaries were calculated from monthly minimum, maximum, mean, and total values using the protocol of [28]. Climatic variables in the main text use WorldClim 2.0 data while additional present climate analyses were trained on present climate data with a coarser spatial grain. Variables included after co-linearity reduction are indicated in bold. (PDF)

**S2 Table. Covariate clusters in primary analyses.** This table reports the identity of covariates in each of two identified co-linear clusters. The central variable in each cluster, with the highest mean correlation with all other variables in the cluster, used to represent the cluster in analyses, appears in bold;the Pearson correlation between each covariate and that central variable is also reported.
(PDF)

**S3 Table. Interaction strength in village-level model.** This table reports the interaction strength of top ranked pair-wise interactions in the boosted regression tree model for infection presence at the village scale.
(PDF)

**S4 Table. Interaction strength in hotspot-level model.** This table reports the interaction strength of top ranked pair-wise interactions in the boosted regression tree model for hotspot identity.
(PDF)

**S5 Table. Covariate clusters in present climate analyses.** This table reports the identity of covariates in each of two identified co-linear clusters. The central variable in each cluster, with the highest mean correlation with all other variables in the cluster, used to represent the cluster in analyses, appears in bold; the Pearson correlation between each covariate and that central variable is also reported.
(PDF)

**S1 Fig. Northern partial dependence plots.** This figure depicts partial dependence plots showing the effect of (a) identity as a fishing village, (b) dog population, (c) number of healthcare supervisor visits, and (d) standard deviation in elevation on probability of parasite presence in northern villages. Histograms represent the distribution of values for these covariates amongst all training villages.
(PDF)

**S2 Fig. Southern partial dependence plots.** Partial dependence plots showing the effect of (a) Cluster 1 (represented by annual precipitation [Bioclim12]), (b) number of healthcare supervisor visits, (c) dog population, and (d) temperature of the driest quarter on probability of parasite presence in southern villages. Histograms represent the distribution of values for these covariates amongst all training villages.
(PDF)

**S3 Fig. Hotspot relative importance estimates in northern and southern villages.** This figure depicts relative importance of covariates in predicting hotspot identity in (a) northern and (b) southern villages.
(PDF)

**S4 Fig. Village-level strongest interaction partial dependence plot.** This figure depicts the strongest pair-wise interaction in the boosted regression tree model for parasite presence, between remotely sensed human population (x-axis) and fishing village identity. The dotted line represents villages designated as fishing villages and the solid line those not designated as fishing villages.
(PDF)

**S5 Fig. Hotspot-level strongest interaction partial dependence plot.** This figure depicts the strongest pair-wise interaction in the boosted regression tree model for village hotspot identity, between cluster 1, represented by annual precipitation (Bioclim 12), and mean temperature of

the coldest quarter (Bioclim 11). In this three-dimensional plot the vertical z-axis represents probability of hotspot identity.
(PDF)

**S6 Fig. Present climate relative importance estimates.** This figure depicts relative importance of covariates in predicting (a) *D. medinensis* presence and (b) hotspot identity. Unlike the main analysis, here models are trained on present climate data with a coarser spatial grain than WorldClim data.
(PDF)

**S7 Fig. Present climate presence-absence partial dependence plot.** This figure depicts partial dependence plots showing the effect of (a) dog population, (b) number of healthcare supervisor visits, (c) fishing village identity, and (d) standard deviation in elevation on probability of parasite presence in all villages when present climate estimates are used rather than World-Clim variables. Histograms represent the distribution of values for these covariates amongst all training villages.
(PDF)

**S8 Fig. Present climate hotspot partial dependence plot.** Partial dependence plots showing the effect of (a) number of healthcare supervisor visits, (b) gridded population density, (c) precipitation of the warmest quarter, and (d) standard deviation in elevation on probability of a village being part of a spatial infection hotspot. Histograms represent the distribution of values for these covariates amongst all training villages.
(PDF)

**S1 Code. Sample analysis and visualization code.** This code displays data-sourcing and processing and conducts analyses and visualizations for this paper. Code is an.Rmd file for use with the knitr and rmarkdown packages in the RStudio software.
(RMD)

**S1 Data. Training village presence-absence data.** Dataset of villages under surveillance. Village data include the five year case count, presence over the five years, and all covariates detailed in S1 Data are split between S15 and S16 to include the training/testing split for BRT modeling.
(CSV)

**S2 Data. Testing Village presence-absence data.** Dataset of villages under surveillance. Village data include the five year case count, presence over the five years, and all covariates detailed in S1 Data are split between S15 and S16 to include the training/testing split for BRT modeling.
(CSV)

**S3 Data. Training village hotspot data.** Dataset of villages under surveillance which were able to be scored as within or outside of a spatial hotspot. These data are included independently of S16 and S17 to allow replication of hotspot analyses both with and without employing the SatScan software. Village data include the five year case count, presence over the five years, presence in a hotspot,and all covariates detailed in S1 Data are split between S17 and S18 to include the training/testing split for BRT modeling.
(CSV)

**S4 Data. Testing village hotspot data.** Dataset of villages under surveillance which were able to be scored as within or outside of a spatial hotspot. These data are included independently of S16 and S17 to allow replication of hotspot analyses both with and without employing the

SatScan software. Village data include the five year case count, presence over the five years, presence in a hotspot,and all covariates detailed in S1 Data are split between S17 and S18 to include the training/testing split for BRT modeling.
(CSV)

## Acknowledgments

We thank the Chadian Ministry of Health as well as all Technical Advisors for collecting and providing surveillance data. We would also like to thank Mark Eberhard, Hubert Zirimwaba-gabo, Elisabeth Chop, Karmen Unterwegner, and The Carter Center for insight into ongoing eradication efforts. http://www.cartercenter.org/donate/corporate-government-foundation-partners/index.html Finally, we thank Sarah Guagliardo and Ashton Griffin for consultation on data management and analysis.

## Author Contributions

**Conceptualization:** Christopher A. Cleveland, Richard J. Hall, Andrew W. Park, Ernesto Ruiz-Tiben, Michael J. Yabsley, Vanessa O. Ezenwa.

**Data curation:** Philip Tchindebet Ouakou, Ernesto Ruiz-Tiben, Adam Weiss.

**Formal analysis:** Robert L. Richards.

**Funding acquisition:** Christopher A. Cleveland, Richard J. Hall, Andrew W. Park, Michael J. Yabsley, Vanessa O. Ezenwa.

**Investigation:** Robert L. Richards, Philip Tchindebet Ouakou, Vanessa O. Ezenwa.

**Methodology:** Robert L. Richards, Andrew W. Park, Vanessa O. Ezenwa.

**Project administration:** Michael J. Yabsley.

**Supervision:** Vanessa O. Ezenwa.

**Writing – original draft:** Robert L. Richards, Vanessa O. Ezenwa.

**Writing – review & editing:** Robert L. Richards, Christopher A. Cleveland, Richard J. Hall, Philip Tchindebet Ouakou, Andrew W. Park, Ernesto Ruiz-Tiben, Adam Weiss, Michael J. Yabsley, Vanessa O. Ezenwa.

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
