## [Decision Letter · Decision Letter 0]

22 Sep 2019

Dear Mr. Richards:

Thank you very much for submitting your manuscript "Identifying drivers of Guinea worm (Dracunculus medinensis) infection in domestic dogs" (#PNTD-D-19-01263) for review by PLOS Neglected Tropical Diseases. Your manuscript was fully evaluated at the editorial level and by independent peer reviewers. The reviewers appreciated the attention to an important problem, but raised some substantial concerns about the manuscript as it currently stands. These issues must be addressed before we would be willing to consider a revised version of your study. We cannot, of course, promise publication at that time.

We therefore ask you to modify the manuscript according to the review recommendations before we can consider your manuscript for acceptance. Your revisions should address the specific points made by each reviewer. 

When you are ready to resubmit, please be prepared to upload the following:

(1) A letter containing a detailed list of your responses to the review comments and a description of the changes you have made in the manuscript.

(2) Two versions of the manuscript: one with either highlights or tracked changes denoting where the text has been changed (uploaded as a "Revised Article with Changes Highlighted" file); the other a clean version (uploaded as the article file).

(3) If available, a striking still image (a new image if one is available or an existing one from within your manuscript). If your manuscript is accepted for publication, this image may be featured on our website. Images should ideally be high resolution, eye-catching, single panel images; where one is available, please use 'add file' at the time of resubmission and select 'striking image' as the file type. 

Please provide a short caption, including credits, uploaded as a separate "Other" file. If your image is from someone other than yourself, please ensure that the artist has read and agreed to the terms and conditions of the Creative Commons Attribution License at http://journals.plos.org/plosntds/s/content-license (NOTE: we cannot publish copyrighted images). 

(4) If applicable, we encourage you to add a list of accession numbers/ID numbers for genes and proteins mentioned in the text (these should be listed as a paragraph at the end of the manuscript). You can supply accession numbers for any database, so long as the database is publicly accessible and stable. Examples include LocusLink and SwissProt.

(5) To enhance the reproducibility of your results, we recommend that you deposit your laboratory protocols in protocols.io, where a protocol can be assigned its own identifier (DOI) such that it can be cited independently in the future. For instructions see http://journals.plos.org/plosntds/s/submission-guidelines#loc-methods

While revising your submission, please upload your figure files to the Preflight Analysis and Conversion Engine (PACE) digital diagnostic tool, https://pacev2.apexcovantage.com/ PACE helps ensure that figures meet PLOS requirements. To use PACE, you must first register as a user. Then, login and navigate to the UPLOAD tab, where you will find detailed instructions on how to use the tool. If you encounter any issues or have any questions when using PACE, please email us at figures@plos.org.

We hope to receive your revised manuscript by Nov 21 2019 11:59PM. If you anticipate any delay in its return, we ask that you let us know the expected resubmission date by replying to this email.

To submit a revision, go to https://www.editorialmanager.com/pntd/ and log in as an Author. You will see a menu item call Submission Needing Revision. You will find your submission record there. 

Sincerely,

Jeremiah M. Ngondi, MB.ChB, MPhil, MFPH, Ph.D

Associate Editor

Banchob Sripa

Deputy Editor

Reviewer's Responses to Questions

**Key Review Criteria Required for Acceptance?**

**Methods**

-Are the objectives of the study clearly articulated with a clear testable hypothesis stated?

-Is the study design appropriate to address the stated objectives?

-Is the population clearly described and appropriate for the hypothesis being tested?

-Is the sample size sufficient to ensure adequate power to address the hypothesis being tested?

-Were correct statistical analysis used to support conclusions?

-Are there concerns about ethical or regulatory requirements being met?

Reviewer #1: L132 ‘We obtained data on D. medinensis incidence in domestic dogs…’

Should the term ‘incidence’ be replaced with ‘cases’? 

If incidence is the correct term, it would be helpful to know more information on how it was calculated given incidence requires knowledge on the population at risk e.g. was the population defined as dogs from the villages studied in this investigation? Given dogs less than 10-14 months old can’t be GW positive, does the population include puppies? Also, is a dog that has multiple worms counted twice? 

L141-142 ‘…but the number of infections per village varied widely (e.g. 2016 range: 0-71), and generally increased over time (Supplemental Fig 1).’

Given dog infections were only realised in ~2012, is this increase due to the increased surveillance effort for dog cases over time? 

What supplementary figure is this referring to? S1 is a table and I can’t work out which figure it would be, perhaps it wasn’t uploaded?

L147-150 ‘Finally, given recent work on the genetic structure of Guinea worm in Chad identifying Northern and Southern parasite sub populations…’

This is a neat idea. However, I can’t find any mention in the methods as to how this division was made? Were clear criteria used to determine this variable for villages? Some clarity on how this was done would be useful.

L151-152 ‘…we collated data on 32 environmental and demographic variables…’

Has there been any assessment for the models being overfitted or for collinearity? 

I would want to see a correlation matrix for the predictor variables, or at least some report of the absence/presence of collinearity. I suspect that there is strong multicollinearity. If so, caution should be taken as even BRT models are not safe from the effects of strong collinearity (see Dormann et al. "Collinearity: a review of methods to deal with it and a simulation study evaluating their performance." Ecography 36.1 (2013): 27-46).

L 156-157 ‘…a single summary value, for the 6-year study period, was included for each village.’

What is the summary value (max, mean, median)?

L165-166 ‘…lower spatial resolution data collected via remote sensing over the same time period as the parasite observations.’

What is the source of this remote sensing data?

L174 ‘Spatial hotspot analysis…’

How was the location of the villages collected, via the GWEP staff?

More importantly, did the spatial analysis require a projection, if so what projection was used? In the Rcode in the supplementary material EPSG:26978 is used. However, this projection is for Kansas south and not Chad. This could lead to distortions that could impact any spatial analysis.

L187-191 ‘…(i) predicting whether D. medinensis was detected in a village over the 6-year study period and (ii) predicting whether…’

This sentence is very helpful as it clearly states the purpose of the BRTs and mentions an important consideration of villages in hotspots. In fact, this whole paragraph for BRT methods is very well written. 

My only criticism is that AUC is not mentioned here and yet it is a key part of the reporting in the results (L224).

L191-194 ‘…we also (i) allowed interactions between longitude/latitude and other covariates…’

(1) Were interactions allowed between collinear predictors?

(2) If surveillance effort changed over the course of the observation period, could this be accounted for in the BRT models? 

(3) There is also the effect of control measures that could influence the status of a village. Again, can the BRT models account for a measure of variation in control strategies used in villages e.g. abate might be more aggressively/frequently applied in some areas than others. 

I realise that 2 & 3 might be hard to include in models, but they could be very important. It would be good to at least acknowledge them as caveats in the discussion.

Reviewer #2: Yes the study desing is appropriate to unearth factors predicting endemicity of Gunea worm infection. The population of villages and corresponding number of dogs was also appropriate. Appropriate discussion is presented as to the limitations that may arise due to sample size. The statistical tests performed on the data was also satisfactory as was the conclusions drawn from them. No ethical concenrs are required for these studies.

Reviewer #3: See below. There are issues with the confounding effects of changes in disease surveillance that are likely happening at the same time as the animal epidemic is increasing. It will be hard, if not impossible, to disentangle these. There are other smaller issues, e.g. how fishing villages are classified.

**Results**

-Does the analysis presented match the analysis plan?

-Are the results clearly and completely presented?

-Are the figures (Tables, Images) of sufficient quality for clarity?

Reviewer #1: L209-217 ‘3.1 Identifying spatial hotspots of infection…’

I find this entire section very confusing. Some of the numbers do not add up (see my comments below) or, if they do, require some explaining. 

In addition, I think some small adjustments could help to better guide the reader through the results. For example, rather than reporting the number of ‘GW positive villages / total villages’ in the methods (L140), it would flow better if put in the results. I would then want to know how many villages were identified in hotspots and, within this, how many were GW positive and GW negative villages. This would help to identify how many GW positive villages have not been identified in hotspots and how many GW negative villages are included in hotspots. It would then be intuitive to provide the same break down for the north and south area. 

If possible I would also like to see the sample size/means ± SE for the data on villages that is used in the BRT models. My suggestion would be to either provide a new table or to add this data to table S1 (with a column for all villages, a second for northern villages and a third for southern villages). Maybe just for the WorldClim data?

L211-212 ‘Overall, 44.2% (703 out of 1592) of villages were identified as belonging to a spatial hotspot (Fig 1)…’

Wasn’t there 2125 villages (reported in the methods L135), where has 1592 come from? 

L212-214 ‘… 53.9% (637 out of 1182) of villages belonged to a hotspot, with fewer clusters in the southern/eastern reaches where only 16.1% (66 out of 410) of villages belonged to a hotspot.’

The totals in the brackets sum to the total mentioned in L211 (1592), but again I’m not sure where this total has come from. Am I right in thinking the 1182 and 410 is the total number of villages studied in the north and south respectively?

L224-225 ‘The BRT model fit data on D. medinensis presence across all villages with a mean AUC of 0.895 in cross-validation and an evaluation AUC of 0.897.’

As mentioned in a previous comment, it would be helpful to introduce the use of AUC in the methods.

L232-235 ‘Fishing village identity was only important in northern villages (as suggested in Fig 3c)…’

Mentioning Fig 3 here is misleading as it could be interpreted as this figure showing data for the analysis on just the northern villages (when it is for the analysis on all villages).

L246-249 ‘…although there were many important interactions, the largest of which was between latitude and mean diurnal temperature range…’ 

It isn’t clear to me what the interaction is between latitude and mean diurnal temperature? I don’t think this is mentioned in the discussion and so it isn’t clear why this is important? Also, why is no comment made on these other important interactions? There are some further interacting variables with longitude, do they not hint as to what is going on in the southern villages?

L254-257 ‘Precipitation of the coldest quarter emerged as important in predicting dog infection at the village scale (10.2% Importance). This change…’

This point would be clearer if the actual change was mentioned. 

Figure 1

This might produce an ugly figure, but if possible it would be great if some/all of the following could be identified in the map:

(1) The 14 spatial hotspots. It would help the reader gauge the appropriateness of the spatial scan and provide visual information on the spatial clusters (i.e. their distribution and variation in size).

(2) The north/south divide

(3) GW positive & negative villages

Figure 2 

Nice plots, just need to add the panel labels (A, B, C, D).

Figure 3

Need to add the panel labels. I would also suggest changing the y axis for the ’probability of infection’ so that they are the same in each plot. 

Are the histograms for the frequency of GW positive villages or for all villages? This should be clearer in the figure description. I assume (given the numbers presented) that this is for all villages. If true, would it not be more informative in the context of these plots to show the frequency of GW positive and negative villages? Currently it is hard to see if the model is a good fit to the data.

S1 Table 

Make sure that the names of variables are consistent with the plots for relative importance e.g. in the table there is 'Distance to nearest permanent water', but this does not appear in the plots for relative importance. I assume that 'RiverDist' is the equivalent variable?

Reviewer #2: Yes the analysis matches the pan and results are clearly presented. I found the figures to be reasonably presented. Figures 3 and 4 could use a bit clearer labels on y axes.

Reviewer #3: See below. Some of the results are interesting and informative, e.g. the interaction between fishing status and longitude and the variation in elevation. I have reservations about the other results.

**Conclusions**

-Are the conclusions supported by the data presented?

-Are the limitations of analysis clearly described?

-Do the authors discuss how these data can be helpful to advance our understanding of the topic under study?

-Is public health relevance addressed?

Reviewer #1: (No Response)

Reviewer #2: Yes to all these points.

Reviewer #3: See below. The conclusions in relation to targeting and control and poorly supported by the analyses.

**Editorial and Data Presentation Modifications?**

Reviewer #1: (No Response)

Reviewer #2: (No Response)

Reviewer #3: L94 – in relation to the classical (typical) life cycle, presumably, the ingestion of copepods in water while drinking applies to humans and to other mammalian hosts, as much as the non-classical cycle mentioned in the next line. 

L97 – there is a “potentially” missing here, to read “potentially allowing them” since it is not clear that they do act in this way to humans or other hosts.

L106 – yes the number of dog infections recorded has increased, though it is not clear that this is due to a genuine increase in incidence or to an improvement in recording.

L135-136. Is surveillance really a ‘daily search’ across >2000 villages? Or rather an elicitation of case reporting, via rewards schemes and a reporting network? This description suggests something much more labour intensive, comprehensive and systematic than the latter.

L142 – Figure S1 and Figure S2 were not included in the review manuscript and were not available for download.

**Summary and General Comments**

Reviewer #1: The authors present a village/regional level analysis of the environmental and anthropogenic predictors of Guinea worm infection in dogs in Chad. The most striking result suggests there are different drivers of infection in northern and southern villages. In the north, ‘fishing’ villages have a higher probability of infection, while environmental variables seem more predictive of cases in the south. These results are important for informing control efforts of an eradication program that has entered its final stages.

Overall, this is a well written manuscript on a very topical and interesting subject. However, I think there are several areas for improvement/clarity that need to be made before the manuscript can be accepted for publication:

(1) The analytical methods are appropriate and articulated clearly, but there is no mention of any checks for model overfitting or collinearity of the predictors. Collinearity could cause problems in the interpretation of the BRT model output. If strong collinearity is discovered and the models need to be altered, the results could be substantially changed given that BRT models are sensitive to even small perturbations of the data input.

(2) There needs to be some presentation of the sample sizes for the variables being used in the BRT models. This would help the reader to get a better understanding of the data being analysed. 

(3) Some of the results do not align with that mentioned in the methods and need to either be corrected or an explanation provided. There is also the scope to provide more summary information that would be useful to the reader.

Reviewer #2: (No Response)

Reviewer #3: The eradication of Guinea worm has progressed well and animals are now a barrier to further progress. The topic of this paper is therefore of considerable interest and is ideally suited to this journal. Understanding the biological drivers of Guinea worm infection/incidence in animals and dogs in particular would be valuable to future control efforts. However, I have several reservations about the specifics of this analysis, which, in my view, limit the utility and relevance of this contribution. 

In their summary, the authors offer the results as guidance for targeting control and surveillance, and point towards elevated risks associated with whether villages are engaged in fishing, larger dog populations, longitude, variation in elevation. They also analyse village location in hotspots. Of the effects reported, variation in elevation and, possibly fishing activity (if this is independent of survey methods and variation in surveillance intensity - see below), are potentially the most biologically informative. However, the other effects might alternatively be framed as recommendations to look for dog infections where there are lots of dogs and, if surveillance effort follows infections, to look for more infections where surveillance has been focused previously. Similarly, the effects of longitude are hard to translate into guidance, other than in relation to fishing, and presence in a hotspot appears compromised by the overwhelming effect of apparently categorising the NW Chari into a single hotspot. I detail concerns about each of these areas in the following points.

There is no mention of variation in surveillance, which presumably has varied spatially and temporally over the duration of the study and since the re-emergence of Guinea worm in Chad. It is important to quantify how surveillance might have changed and to take this into account. A glance at the detail of Guinea worm surveillance activities in the “Guinea worm wrap-up” suggests that surveillance has been stepped up across the board in Chad: “Some of the increase in infected dogs reported probably resulted from expansion of the number of villages under active surveillance (VAS) from 1,895 at the end of 2018 to 2,138 as of May 2019.” Villages have also apparently been classified as Level 1 to Level 3 villages, presumably reflecting varying surveillance effort. Which villages were subject to the greatest surveillance and how did this vary over time as both the apparent epidemic and surveillance were growing? Are the authors actually demonstrating the effects of an increase, or spatial intensification, in surveillance effort. For control purposes, this variation in effort might very sensibly be focused on big villages, fishing villages, those in particular regions, or in infection hotspots (as in Levels 1-3). It is important to isolate and account for these obvious and influential logistical, practical changes in surveillance as a driver of variation in the detection of infections, as much as genuine biological risk factors. Either way, these potential effects cannot reliably be described as drivers of infection in an epidemiological sense.

The effect of dog population size on the classification of a village as an “infected village” is not at all surprising, and would be apparent if the incidence of infection was genuinely uniform across all areas. Indeed, it would be surprising if this were not an effect. This “per village” rate of infection, is not to be confused with an effect of dog population size on the “per dog” rate of infection, though the latter has not been quantified or tested. This is likely to be the unavoidable consequence of being more likely to find a dog infection when there are more dogs to look at. The authors acknowledge this at L276. However, it would clearly be a mistake to interpret this as a driver of infection and/or to target surveillance or control on this basis. The distribution of dog population sizes in Figure 3 suggests that almost all villages have small dog populations, perhaps <~100 dogs and so many small villages would still harbor dog infections if surveillance was focused upon the few, larger villages. 

Similarly, the longitude effect, which is the largest single effect across the paper, is not clearly spelled out and so the biology and targeting consequences remain unclear. According to Figure 1, the Chari River runs N-S, but the southern affected area lies to the E and the Northern Area to the W. If we look at Figure 1, this maps out villages in hotspots and those that are not, but not the hotspots themselves. There are no green dots north of about half-way up the country. So are all the orange dots in the North, i.e. all the infected villages in the north, all in one big hotspot, and the others all in very much smaller separate hotspots? The text suggests there are 14, but 14 are not apparent from the figure. The Figure is also cut off at the bottom and would benefit from lines of longitude. Presentational issues aside, the differing magnitude of these hotspots seems likely to introduce a heavy influence of this single large NW hotspot in later analyses. Thus, the analysis reported at L250-257 and in Figure 4 appears largely to distinguish the location of the “NW big hotspot”, i.e. all the villages in the NW, to be in the NW hotspot. Again, this does not appear helpful and would be misleading in targeting. 

L159-162 It is not made clear how villages were classified as a “fishing village” or otherwise, or how the “majority of the population” was determined? According to the supplementary information, this was done by the Guinea worm survey field teams, it is therefore possible that this introduced a systematic bias with respect to Guinea worm incidence. This needs to be shown to be otherwise. That said, the interaction between fishing village classification and location in the NW or SE is potentially of interest.

L262 it is not elevation, but standard deviation in elevation that was influential. This is potentially an important distinction, given the variation in relief associated with surface water availability. 

Incidentally, the analyses appear not to distinguish between villages in which one infection has been recorded and those in which infection is a more frequent occurrence? It appears as though the analysis distinguishes between a binary classification. This seems like a missed opportunity, given the availability of data and the very wide variation in reported cases over the study.

PLOS authors have the option to publish the peer review history of their article (what does this mean?). If published, this will include your full peer review and any attached files.

Reviewer #1: Yes: Jared K Wilson-Aggarwal

Reviewer #2: No

Reviewer #3: No

---

## [Decision Letter · Decision Letter 1]

17 Feb 2020

Dear Mr. Richards,

Thank you very much for submitting your manuscript "Identifying drivers of Guinea worm (Dracunculus medinensis) infection in domestic dogs" for consideration at PLOS Neglected Tropical Diseases. As with all papers reviewed by the journal, your manuscript was reviewed by members of the editorial board and by several independent reviewers. In light of the reviews (below this email), we would like to invite the resubmission of a significantly-revised version that takes into account the reviewers' comments. 

We cannot make any decision about publication until we have seen the revised manuscript and your response to the reviewers' comments. Your revised manuscript is also likely to be sent to reviewers for further evaluation.

Sincerely,

Jeremiah M. Ngondi, MB.ChB, MPhil, MFPH, Ph.D

Associate Editor

Banchob Sripa

Deputy Editor

Reviewer's Responses to Questions

**Key Review Criteria Required for Acceptance?**

**Methods**

-Are the objectives of the study clearly articulated with a clear testable hypothesis stated?

-Is the study design appropriate to address the stated objectives?

-Is the population clearly described and appropriate for the hypothesis being tested?

-Is the sample size sufficient to ensure adequate power to address the hypothesis being tested?

-Were correct statistical analysis used to support conclusions?

-Are there concerns about ethical or regulatory requirements being met?

Reviewer #1: L161 – 164 ‘…we collated data on 35 variables that might contribute to transmission….’

There are still some inconsistencies and areas for clarification in the use of language in the methods, Table S1 and Figure 2. See my specific comments for Table S1 and Figure 2 below. 

An example is ‘RemotePop’ which appears in Figure 2. From the results it looks like this is a measure of the population from remote sensing data? However, there is no mention of this variable (or how it is calculated) in the methods, and it is not in Table S1. 

Aren’t there more than 35 variables considered when you include the controls for surveillance effort? 

L172 ‘Villages were identified by GWEP staff as “fishing” if the majority of families...’

Be more explicit, what is meant by majority, >50%? 

L173 - 176 ‘To control for spatial variation in surveillance, we also included two measures of surveillance effort…’

Add a few words to explain the differences in what these control for, it is not immediately clear. Perhaps just add a line to the same effect of that in your reply to reviewer comments ‘The second variable encompasses some degree of spatio-temporal variation in surveillance as villages have no visits for years that they went unsurveilled.’

L211 – 213 ‘…we diagnosed significant co-linearity in predictors…’

Did you include the variables for sampling effort in the co-linearity analysis? It would be interesting to know whether or not measures of sampling effort are correlated with demographic parameters. 

L213 – 214 ‘Co-linear clusters were reduced to the single most central variable…’

My understanding is that this method involves omitting variables identified in a cluster (except the central variable) from the model. With this in mind, why do variables in the clusters (that are not the central variable) appear in the relative importance plots and interaction tables? For example BIO9, BIO11 and remotely sensed population. Is this because you only omitted variables in a cluster if they went above a threshold e.g. r > 0.7?

L224 – 227 ‘…we also (i) allowed interactions between longitude/latitude and other covariates…’

Why are interactions added into the models when they are ignored in the results and discussion? Furthermore, it isn’t entirely obvious to me how to interpret the interaction results reported in the supplementary tables and figures. I would expect that these interactions are quite important to our understanding and interpretation of the results and therefore some elaboration is required.

Also, I interpreted this to mean that separate N/S analyses were done for both the village and hotspot scales. However, this does not seem to be the case. I would suggest either adding the N/S analyses for the hotspot scale or to be more explicit and to say that this analysis was only conducted for the village scale.

Reviewer #3: In relation to objectives and design:

I think that in this revision there remains a mismatch between the prominently stated ambition to identify drivers of infection in dogs and the reality, which is identification of correlates of village-level detection of dog infections and a coarse analysis of locations of infection hotspots. I don't think this is being fussy. The authors just need to be a bit more honest and conservative in the headline elements of the paper when it comes to talking about what they are able to do with the data they have. Therefore, the Methods adopted are appropriate to a different aim of understanding correlative patterns in the long-term detection of dog infections in villages, but they are not suitable, and the data underpinning the analyses are not suitable, for the stated aim of an analysis of causative drivers of infection in dogs. I don't think therefore that it is possible to support the title or Abstract with the data used and Methods adopted with any amount of revision of the approach. Rather the stated ambition, title and scope of the paper needs to shift to fit with the data, Methods and analyses.

I have no concerns about ethical or regulatory requirements.

**Results**

-Does the analysis presented match the analysis plan?

-Are the results clearly and completely presented?

-Are the figures (Tables, Images) of sufficient quality for clarity?

Reviewer #1: L292 – 295 ‘The most important variables included cluster 1 (represented by Bioclim 12, 36.0% Importance)… mean temperature of the coldest quarter (10.2% Importance) and mean temperature of the driest quarter (Bioclim 9, 9.6% Importance).’

To help the reader there needs to be some consistency when referring to Bioclim variables. I would suggest using the format you use at the end of this sentence when referring to Bioclim9 i.e. say what it actually is and add the shorthand in brackets.

Figure 1

Can you provide a reason for why the spatial hotspots overlap? This doesn’t seem right to me and I would suggest that something has gone wrong, however, I do not know the intricacies of the spatial scan and whether or not this is normal behaviour. The obvious answer would be due to temporal overlap, but this cannot be the case in this analysis as GW positive villages are identified using the entire observation period. Is there something in the R package guidance or literature that could explain this, or is this an error? Perhaps the scan is using the temporal aspects of the data rather than ignoring it?

Table S1

A few suggestions to help guide the reader:

1) Include the abbreviations used in Figure 2 in brackets or add them to the caption for Figure 2.

2) Add RemotePop

3) Add a section of rows for the variables controlling for sampling effort and report their mean & SE

4) For the last three column headings keep the ‘N=’ but change the rest of the text to something like ‘All villages’, ‘Northern villages’ and ‘Southern villages’. Then, in the legend, put something to the effect of ‘Where appropriate the means (± SE) are reported’. 

5) Include the number and/or proportion of villages that were identified as fishing villages, rather than leaving it blank.

6) My understanding of the choice of collinearity analysis is that many of these covariates were omitted from the model. If so, consider highlighting those that were used in the model by putting them in bold.

Table S2

This is not referenced in the main text. Furthermore the relevance of these interactions and how to interpret them should be discussed, currently it is not clear why they were considered in the first place or what they mean. 

Captions and legends

Check these carefully as some are not correct e.g. for Figure S7 it mentions panels a-d, but only two graphs are presented.

Reviewer #3: (No Response)

**Conclusions**

-Are the conclusions supported by the data presented?

-Are the limitations of analysis clearly described?

-Do the authors discuss how these data can be helpful to advance our understanding of the topic under study?

-Is public health relevance addressed?

Reviewer #1: L320 – 322 ‘Overall, our results suggest that…’

The final part of this sentence is a bit repetitive of the previous and doesn’t really add much. How can correlates with the broad hotspot scale help the GWEP? 

L383 – 402 ‘In addition to our village level analyses, the identification of spatial hotspots…’

I think this paragraph discussing the hotspot analysis could be improved. While the points raised are interesting, I think more needs to be said about the variables that have been identified to characterise a hotspot and how these can be used by the GWEP. The model suggests that hotspots might be predicted/characterised by the mean annual precipitation, and the correlation of this variable with a number of other variables is worth discussion. There were also interactions that were identified, however, as I mentioned earlier, many of these are with variables in the same collinear cluster and will not be present once removed from the model.

Why was the hotspot analysis not separated into N/S as with the village level analysis? 

L417 – 419 ‘…suggesting that they served as an effective control.’

I initially understood this to mean effective control for GW, rather than biases in the sampling of infections. I would suggest rewording this sentence to make it clear that you are referring to sampling bias.

Reviewer #3: In relation to the take-home messages in the Abstract, please see my comments above and below.

More specifically, given that health-care worker visitation was a key factor in detection of infections, I would not consider this an "intrinsic" (or demographic, as in the previous paragraph) characteristic of the village. 

Climate variables are somewhat helpful as is elevation, but human population size and health worker visitation rate are the main contributors to the models' explanatory power. I think that when it comes to the models being "highly predictive" a little more care should be taken to relate the predictability to which factors are epidemiologically informative. There remain some assertive phrases, e.g. line 320-1, about how this might be used in targeting control efforts that I am not sure are strongly supported.

**Editorial and Data Presentation Modifications?**

Reviewer #1: (No Response)

Reviewer #3: Thanks to the authors for having dealt with, or acknowledged and responded positively too, many of the substantive concerns in my initial review.

Nevertheless, I find I have a large number of presentational and editorial points. I think the paper would still be considerably better if these were addressed. I have tried not to be too particular, but some of these are quite important points. These primarily relate to the points raised above about being a bit more conservative in relation to the scope and outcomes of the study.

Title

Drivers: Given all the caveats that are now more thoroughly explored, it is probably not justified to use "Drivers" in the title. It might be better to use "Correlates of" or "Patterns in"

Infection in dogs: The analyses would better be a description of "villages in which dog infections are recorded" or some more accurate way of conveying the subject of the analysis, which is not of dog infections, but of recording detection of infections at a village level over an extended period.

Abstract

Guinea in Guinea worm is usually capitalised, as it derives from the country. (e.g. Line 38).

Overall, I found the wording of the first Background paragraph of the Abstract to be problematic and imprecise. For example: It is not necessarily the case that the epidemiology changed, or that this happened "at the same time". Why is 2012 selected? Domestic dogs were observed to be hosts to Guinea worm multiple times and in multiple countries before 2012 and outside of Chad, and before they were found in Chad. It is not the dogs that challenge efforts exactly, they are indifferent, but the presence of infection in a non-human host creates a challenge for those charged with eradication. See my point above about "driving" and drivers. Although these would be good to know, this study does not, in the end, reveal them. I am sorry to be picky about these finer points, but attention to this sort of detail will help with the tone of the rest of the paper. 

Might it be better throughout to talk about the "detection of infection" rather than "infection" per se? (e.g. Line 47)

Similarly, it is not "Guinea worm parasite presence" but "detection of Guinea worm infection in a village" (Line 54).

I would suggest that n healthcare visitors is not "intrinsic to the village", since it is increased or decreased by the eradication program. (Line 54)

I would also suggest that in terms of "landscape scale ecology" the insight is limited, since the relevant factors were not especially ecological. But this is a subjective point. (Line 56)

**Summary and General Comments**

Reviewer #1: The authors have made appropriate responses to comments on the previous version of this manuscript, and many of the issues and concerns have been addressed. Overall, this is a well written manuscript and this body of work has the potential to significantly contribute to our understanding of a zoonotic disease that is on the brink of eradication. However, the methodology and results newly presented by the authors still raise considerable questions, and there are several areas for clarification/improvements that need to be addressed before the manuscript can be accepted for publication:

(1) I have several concerns about the hotspot analysis. While figure 1 is much improved and helps the reader see the results of the spatial scan analysis, it is surprising to see that there is considerable overlap of hotspots. This suggests to me that there is an error as I would only expect overlaps if the data were temporal in nature. This either needs rectifying or requires some explanation in the methods. 

(2) The authors have appropriately adapted their analysis to consider collinearity however, there are some discrepancies that could require a re-run of the analysis. ‘Co-linear clusters were reduced to the single most central variable’, implying that the non-central variables were removed from the model, but this does not seem to be the case. This needs clarification as to if a different method was used or, alternatively, the models will need to be corrected and run again. 

(3) Interactions are included in models, but they are overlooked in the results and discussion. The interactions between variables will be important for the interpretation of the results and should be given more attention. Currently it is not clear why they are included in the model or how they should be interpreted.

Reviewer #3: As indicated above, I feel that although the response document acknowledge much of the earlier critique, the revisions have not sufficiently addressed the issue of scope and ambition of the analyses. In essence, this is an analysis of correlates of the village-level detection of infection over an extended period. It is not an analysis of the drivers of dog infections. Some of this change of tone is in the main text, but the Title, Abstract and Author Summary remain too assertive.

PLOS authors have the option to publish the peer review history of their article (what does this mean?). If published, this will include your full peer review and any attached files.

Reviewer #1: Yes: Jared Wilson-Aggarwal

Reviewer #3: No
---

## [Decision Letter · Decision Letter 2]

26 May 2020

Dear Mr. Richards,

Thank you very much for submitting your manuscript "Identifying correlates of Guinea worm (Dracunculus medinensis) infection in domestic dog populations" for consideration at PLOS Neglected Tropical Diseases. As with all papers reviewed by the journal, your manuscript was reviewed by members of the editorial board and by several independent reviewers. The reviewers appreciated the attention to an important topic. Based on the reviews, we are likely to accept this manuscript for publication, providing that you modify the manuscript according to the review recommendations. 

Sincerely,

Jeremiah M. Ngondi, MB.ChB, MPhil, MFPH, Ph.D

Associate Editor

Banchob Sripa

Deputy Editor

Editorial comments: Data Availability

As per PLOS data policy (https://journals.plos.org/plosntds/s/data-availability) the reviewers have raised concerns with the data availability statement that you have provided. I have noted that there is a recent publication on a similar subject and locations from Chad https://journals.plos.org/plosntds/article?id=10.1371/journal.pntd.0008170 that has made the datasets publicly available as per the PLOS data policy. Notably, Dr. Tchonfienet Moundai is a co-author on this recent paper. I therefore suggest that you consult Dr. Tchonfienet Moundai to cross-check if the minimum data sets for this paper can be made available to the public as per the PLOS data policy. 

Reviewer's Comments

**Editorial and Data Presentation Modifications?**

Reviewer #1: (No Response)

Reviewer #3: For the avoidance of doubt, none of these points is substantive and all are easily dealt with, with no further recourse to the reviewers. 

L77 - This suggests that eradication might have been achieved in the 1980s, whereas the target of eradication was set in the 1980s.

L84 - I would say that it is not so much the annual occurrence as the persistence of human cases.

L85 - Should you add the species name for dogs?

L91-3 - Recent work has hypothesised this pathway, and identified some contributing elements, without demonstrating it.

L138-9 and L259-50 - Clarify village-level prevalence, as this could still be misconstrued as the prevalence among dogs.

L139 and L250 - Clarify, this is dog infections and whether this is separate infections, or dogs with worms, as many dogs have >1 worm infection.

L322 - "Canine Guinea worm" implies that it is a dog parasite, better to say Guinea worm infection in dogs

L357 - It is not higher risk of Guinea worm infection in dogs, but greater likelihood of detecting dog infections in villages.

L439 - Clarify that there is no transmission between dogs and humans, as such.

**Summary and General Comments**

Reviewer #1: The authors have made appropriate changes to the manuscript and provided clarification on the methods used. I have no further suggestions or queries and think this manuscript is acceptable for publication. I would like to congratulate the authors on producing a well written manuscript with interesting results that will help guide future research towards the control of Guinea worm disease.

Reviewer #3: The authors have tackled my last set of suggestions and recommendations in good part,

Some minor points that are definitely optional but which might improve understanding and clarity are in the box above. None of these requires any further review or assessment. 

One easy point that does require correction, at L326, and which was raised previously is that it is not elevation that was found to be influential, but standard deviation in elevation.

I have no other concerns.

PLOS authors have the option to publish the peer review history of their article (what does this mean?). If published, this will include your full peer review and any attached files.

Reviewer #1: Yes: Jared Wilson-Aggarwal

Reviewer #3: No
---

## [Editor Report · Decision Letter 3]

20 Jul 2020

Dear Mr. Richards,

We are pleased to inform you that your manuscript 'Identifying correlates of Guinea worm (Dracunculus medinensis) infection in domestic dog populations' has been provisionally accepted for publication in PLOS Neglected Tropical Diseases.

Best regards,

Jeremiah M. Ngondi, MB.ChB, MPhil, MFPH, Ph.D

Associate Editor

Banchob Sripa

Deputy Editor

---

## [Editor Report · Acceptance letter]

3 Sep 2020

Dear Mr. Richards,

We are delighted to inform you that your manuscript, "Identifying correlates of Guinea worm (Dracunculus medinensis) infection in domestic dog populations," has been formally accepted for publication in PLOS Neglected Tropical Diseases.

Best regards,

Shaden Kamhawi

co-Editor-in-Chief

Paul Brindley

co-Editor-in-Chief
